# A Preliminary Investigation towards the Risk Stratification of Allogeneic Stem Cell Recipients with Respect to the Potential for Development of GVHD via Their Pre-Transplant Plasma Lipid and Metabolic Signature

**DOI:** 10.3390/cancers11081051

**Published:** 2019-07-25

**Authors:** Daniel Contaifer, Catherine H. Roberts, Naren Gajenthra Kumar, Ramesh Natarajan, Bernard J. Fisher, Kevin Leslie, Jason Reed, Amir A. Toor, Dayanjan S. Wijesinghe

**Affiliations:** 1Department of Pharmacotherapy and Outcomes Sciences, School of Pharmacy, Virginia Commonwealth University, Richmond, VA 23298, USA; 2Department of Internal Medicine, School of Medicine, Virginia Commonwealth University, Richmond, VA 23298, USA; 3Department of Microbiology, School of Medicine, Virginia Commonwealth University, Richmond, VA 23298, USA; 4Department of Physics, College of Humanities and Sciences, Virginia Commonwealth University, Richmond, VA 23284, USA; 5Massey Cancer Center, Virginia Commonwealth University, Richmond, VA 23298, USA; 6Institute for Structural Biology Drug Discovery and Development (ISB3D), VCU School of Pharmacy, Richmond, VA 23298, USA; 7Da Vinci Center, Virginia Commonwealth University, Richmond, VA 23284, USA

**Keywords:** stem cell transplantation, graft vs. host disease, risk stratification, metabolomics, lipidomics

## Abstract

The clinical outcome of allogeneic hematopoietic stem cell transplantation (SCT) may be influenced by the metabolic status of the recipient following conditioning, which in turn may enable risk stratification with respect to the development of transplant-associated complications such as graft vs. host disease (GVHD). To better understand the impact of the metabolic profile of transplant recipients on post-transplant alloreactivity, we investigated the metabolic signature of 14 patients undergoing myeloablative conditioning followed by either human leukocyte antigen (HLA)-matched related or unrelated donor SCT, or autologous SCT. Blood samples were taken following conditioning and prior to transplant on day 0 and the plasma was comprehensively characterized with respect to its lipidome and metabolome via liquid chromatography/mass spectrometry (LCMS) and gas chromatography/mass spectrometry (GCMS). A pro-inflammatory metabolic profile was observed in patients who eventually developed GVHD. Five potential pre-transplant biomarkers, 2-aminobutyric acid, 1-monopalmitin, diacylglycerols (DG 38:5, DG 38:6), and fatty acid FA 20:1 demonstrated high sensitivity and specificity towards predicting post-transplant GVHD. The resulting predictive model demonstrated an estimated predictive accuracy of risk stratification of 100%, with area under the curve of the ROC of 0.995. The likelihood ratio of 1-monopalmitin (infinity), DG 38:5 (6.0), and DG 38:6 (6.0) also demonstrated that a patient with a positive test result for these biomarkers following conditioning and prior to transplant will be at risk of developing GVHD. Collectively, the data suggest the possibility that pre-transplant metabolic signature may be used for risk stratification of SCT recipients with respect to development of alloreactivity.

## 1. Introduction

Transplantation of hematopoietic progenitors from an HLA-matched donor is a curative procedure for many patients with hematologic malignancies and disorder of hematopoiesis. Graft vs. host disease (GVHD) is a frequently observed complication of stem cell transplantation (SCT), which contributes to transplant-related mortality and adversely impacts clinical outcomes following transplantation. GVHD after allogeneic hematopoietic stem cell transplantation (HSCT) is a reaction of donor immune cells to recipient tissues. An inflammatory cascade triggered by the preparative regimen causes activated donor T cells to target and destroy epithelial cells. About 35–50% of HSCT recipients will develop acute GVHD. It is mediated by donor-derived T cells responding to minor histocompatibility antigens encountered in the recipient. The T cells encounter these alloantigens, undergo activation, and perform functions such as cytokine secretion (IL-2, IL-4, IL-10, IL-12, IL-17, and interferon gamma by helper T cell subsets) and target lysis (granzyme and perforin secretion by cytotoxic T cells). These functions are accompanied by significant metabolic adaptations in the T cells, including increased glycolysis and oxygen consumption as well as cytokine production [1]. As an example, higher levels of GLUT 1 (glucose transporter 1) expression have been observed in activated T cells, suggesting increased metabolic and biosynthetic rates [2]. Supporting this, correlation has been shown between intracellular ATP concentration in T cells and severity of clinical GVHD in humans, and between increasing glycolysis and GVHD in murine models [3,4]. These observations suggest that donor T cell activation and consequent metabolic and biosynthetic changes may correlate with clinical events as immune reconstitution occurs following SCT, and alloreactivity is triggered.

Just as metabolic changes in the T cell are crucial to the onset of immune reactions, the metabolic milieu in which the T cells find themselves influences their function. In this respect, the lipid molecules constitute a family of important functional mediators. The effects of some lipid molecules on the T cells have been studied recently. These effects include lysophosphatidylserine (lysoPS)-mediated suppression of IL-2 production and suppression of T cell proliferation [5]. This effect is mediated via LPS3/G protein coupled receptor 174, which triggers IL-2 production in CD4+ T cell. Another enzyme, acid sphingomyelinase (ASMase), generates ceramide and modulates signaling cascades involving CD3 and CD28. It is involved in Th1 and Th17 responses through its effect on signal transducer and activator of transcription 3 (STAT 3) and the mammalian target of rapamycin (mTOR) [6]. An acid sphingomyelinase deficient mouse model experiences attenuation of GVHD [7]. Along the same lines, leukotriene C4 has been shown to be important for airway inflammation when administered to murine models along with IL33 [8]. Consistent with such results are the observations that T cells in acetyl-CoA carboxylase deficient mice are resistant to induction of GVHD [9]. T effector cells have been shown to increase their reliance on fatty acid metabolism during GVHD as well [10,11], for example, Prostaglandin E2 (PGE_2_) has been implicated in modulating T cell effects of mesenchymal stromal cells on T cell populations [12] and PGE_2_ priming of T cells reduces GVHD when localized to the site of alloreactivity [13]. Further, bone marrow stromal cells also exert an ameliorating influence on GVHD through indoleamine 2,3-dioxygenase (IDO) and PGE_2_ expression [14]. These lipid-mediated effects have been targeted in the treatment of GVHD, for instance, the effectiveness of leukotriene inhibitor monteleukast has long been recognized in managing GVHD of the lung [15,16]. Additionally, prostaglandins have been studied in GVHD prevention strategies [17], particularly PGE_2_ [13]. A leukotriene inhibitor, eicosapentanoic acid (EPA), has also been studied in the treatment of GVHD as well as prophylaxis [18,19]. Prostaglandin mediators FT1050 (16,16-dimethyl PGE2) and FT4145 (dexamethasone) are also being studied in clinical trials of GVHD prophylaxis using ex-vivo modification of the allograft [20,21]. These observations make it crucial to gain an understanding regarding the lipid and metabolic changes that come about following dose-intensive myeloablative conditioning, and how the ensuing metabolome and the lipidome might impact T cell function following SCT. Modern methods of lipidomics and metabolomics allow us to study such changes in detail [22,23,24,25,26]. In this paper, we describe the lipidomic and metabolomic profiles of patients undergoing myeloablative conditioning and stem cell transplantation and try to understand the role of these metabolites in mediating alloreactivity, and for potentially predicting GVHD.

## 2. Materials and Methods

### 2.1. Patients

Patients were enrolled prospectively in a Virginia Commonwealth University (VCU) Institutional Review Board (IRB)-approved observational pilot study; patients provided written informed consent prior to enrollment (Ethical code: 45 CFR 46.108(b) and 45 CFR 46.109(e) and 45 CFR 46.110 by VCU IRB Panel A, permission date: 27 March 2019). Patients underwent myeloablative conditioning followed by either HLA-matched related (MRD), or HLA-matched unrelated donor (URD); patients undergoing autologous stem cell transplantation (auto) were also included as controls (Table 1). HLA matching was at the allelic level; allogeneic HCT recipients received ATG, calcineurin inhibitors, and either mycophenolate mofetil or methotrexate for GVHD prophylaxis. Blood samples were drawn after completion of myeloablative conditioning therapy on day 0 prior to SCT. Blood was processed for plasma collection and frozen at −80 °C until mass spectroscopy-based metabolomic and lipidomic analysis. Given the small patient cohort, acute and chronic GVHD data were pooled, and Glucksberg and NIH Consensus criteria were used to diagnose and stage GVHD.

### 2.2. Lipid and Metabolite Extraction for LC-MS/MS Analyses

Blood plasma lipids extraction was carried out using a biphasic solvent system of cold methanol, methyl tertiary butyl ether (MTBE), and water with some modifications (Matyash et al., 2008). In detail, 225 µL of cold methanol containing a mixture of odd chain and deuterated lipid internal standards [lysoPE(17:1), lysoPC(17:0), PC(12:0/13:0), PE(17:0/17:0), PG(17:0/17:0), sphingosine (d17:1), d7 cholesterol, SM(17:0), C17 ceramide, d3 palmitic acid, MG(17:0/0:0/0:0), DG(18:1/2:0/0:0), DG(12:0/12:0/0:0), and d5 TG(17:0/17:1/17:0)] were added to a 20 µL blood plasma aliquot in a 1.5 mL polypropylene tube and then vortexed. Next, 750 µL of cold MTBE was added, followed by vortexing and shaking with an orbital mixer. Phase separation was induced by adding 188 µL of MS-grade water. Upon vortexing (20 s), the sample was centrifuged at 12,300 rpm for 2 min. The upper organic phase was collected in two 300 µL aliquots and evaporated with a vapor trap. Dried extracts were resuspended using 110 µL of a methanol/toluene (9:1, *v*/*v*) mixture containing CUDA (50 ng/mL; internal standard for quality control of injection) with support of vortexing (10 s), and centrifuged at 800 rpm for 5 min, followed by transferring 100 µL of the supernatant into auto-sampler vial with an insert. The lower polar layer was collected and an aliquot of 125 µL was evaporated to dryness in a SpeedVac, and resuspended in acetonitrile for polar metabolite analysis via HILIC LC-MS/MS method.

### 2.3. Metabolomics: GC-MS Metabolite Extraction

A total of 30 μL of plasma sample was added to a 1.0 mL of pre-chilled (−20 °C) extraction solution composed of acetonitrile, isopropanol, and water (3:3:2, *v*/*v*/*v*). Samples were vortexed and shaken for 5 min at 4 °C using the orbital mixing chilling/heating plate. Next, the mixture was centrifuged for 2 min at 14,000 rcf. Then, 450 μL of the supernatant was dried with cold trap concentrator. The dried aliquot was then reconstituted with 450 μL acetonitrile:water (50:50) solution and centrifuged for 2 min at 14,000 rcf. The supernatant was transferred to a polypropylene tube, and subjected to drying in a cold trap. The process of derivatization began with the addition of 10 μL of 40 mg/mL Methoxyamine hydrochloride solution to each dried sample and standard. Samples were shaken at maximum speed at 30 °C for 1.5 h. Then, 91 μL of MSTFA + FAME mixture was added to each sample and standard, and capped immediately. After shaking at maximum speed at 37 °C, the content was transferred to glass vials with micro-inserts inserted and capped immediately.

### 2.4. Lipids: LC-MC/MC Conditions

Untargeted lipid analysis was undertaken with Sciex TripleTOF 6600 coupled to Agilent 1290 LC. Lipids were separated on an Acquity UPLC CSH C18 column (100 × 2.1 mm; 1.7 µm) (Waters, Milford, MA, USA). The column was maintained at 65 °C and the flow-rate of 0.6 mL/min. The mobile phases consisted of (A) 60:40 (*v*/*v*) acetonitrile:water with 10 mM ammonium acetate and (B) 90:10 (*v*/*v*) isopropanol:acetonitrile with 10 mM ammonium acetate. The separation was conducted following a stepwise gradient: 0–2 min 15–30% (B), 2–2.5 min 48% (B), 2.5–11 min 82% (B), 11–11.5 min 99% (B), 11.5–12 min 99% (B), 12–12.1 15% (B), and 12–14 min 15% (B). Negative and positive electrospray ionization (ESI) modes were applied with nitrogen serving as the desolvation gas and the collision gas. The parameters for detection of lipids were as follows: Curtain Gas: 35; CAD: High; Ion Spray Voltage: 4500 V; Source Temperature: 350 °C; Gas 1: 60; Gas 2: 60; Declustering Potential: +/−80 V, and collision energies +/−10.

### 2.5. Metabolites: GC-MS Conditions

A Leco Pegasus IV time of flight mass spectrometer coupled with Agilent 6890 GC equipped with a Gerstel automatic liner exchange system (ALEX) that included a multipurpose sample (MPS2) dual rail, and a Gerstel CIS cold injection system (Gerstel, Muehlheim, Germany) was used to complement HILIC metabolite analysis. The transfer line was maintained at 280 °C. Chromatography separation was achieved on a 30 m long, 0.25 mm i.d. Rtx-5Sil MS column (0.25 μm 95% dimethyl 5% diphenyl polysiloxane film) with the addition of a 10 m integrated guard column was used (Restek, Bellefonte, PA, USA) with helium (99.999%; Airgas, Radnor, PA, USA) at a constant flow of 1 mL/min. The oven temperature was held constant at 50 °C for 1 min and then ramped at 20 °C/min to 330 °C at which it was held constant for 5 min. The GC temperature program was set as follows: 50 °C to 275 °C final temperature at a rate of 12 °C/s and hold for 3 min. The injection volume was 1 μL in splitless mode at 250 °C. Electron impact ionization at 70 V was employed with an ion source temperature of 250 °C. The scan mass ranged from 85 to 500 Da with an acquisition rate of 17 spectra/second.

### 2.6. Statistical Analysis

Prior to statistical analysis, metabolomic and lipidomic data were subjected to preprocessing as follows. Data were first normalized by a variant of a ‘vector normalization’ by calculating the sum of all peak heights for all identified metabolites for each sample and thereafter normalizing each compound by the total average of the sum. Multivariate statistical analysis tends to focus on metabolites with high intensities. To avoid this tendency, log scaling was applied to reduce the effect of large peaks and scale the data into a more normally distributed pattern. Pareto scaling, which uses the square root of the standard deviation as the scaling to change the emphasis from metabolites with high concentrations to those with moderate or small abundances, was also used for analyzing parameters with large variation.

The patient cohort was divided into patients that developed either acute or chronic GVHD and patients that did not develop GVHD. Because of the small sample size and since the analysis was performed before transplantation, autologous stem cell transplant patients that did not develop GVHD were grouped with allograft recipients that also did not develop GVHD. Partial least squares discriminant analysis (PLS-DA) was used to create a bilinear model to fit the data (Figure 1). Multivariate statistical methods such as PLS-DA have been introduced to reduce the complexity of metabolic spectra and help identify meaningful patterns in high-resolution mass spectrometric data. In this method, the PLS-DA scores can be filtered through calculation of the variable importance in projection (VIP), and used to estimate the contribution of lipids and metabolites for class separation. The top 20 most important variables in the PLS-DA model were selected for further investigation (Figure 1).

Because of the relatively small sample size, cross validation was employed to further evaluate the classification model performance. An algorithm based on support vector machine (SVM) was used to identify the top 20 variable importance in projection (VIP) to further select the best hyperplane that represents the largest separation between the two groups (Figure 1). This method was coupled with Monte-Carlo cross validation (MCCV) through balanced subsampling to support the validation in the small sample cohort (Figure 1). In each MCCV, two thirds of the samples are used to evaluate the feature importance and the remaining one-third is used as test population. The SVM and MCCV allows a multivariate area under the curve of the receiver operating characteristic (AUCROC) analyses to estimate the success of the classification model (Figure 1), creating several AUCROC models to test performances with different numbers of predictors. This procedure was repeated multiple times to calculate the performance estimates and build a confidence interval for each model. Based on the performance in the multivariate AUCROC, the best number of predictors is a reference for selection of potential biomarkers [27].

Univariate AUCROC analyses was used to find potential biomarkers with sufficient power to separate the groups (Figure 1). The result shows the *t*-test *p*-value and the AUCROC value with the confidence interval computed using 500 bootstrap replications. The criteria to choose the stronger potential biomarkers for GVHD was to select compounds with the highest AUCROC performance and lowest *p*-value (*p* < 0.05). The calculated optimal cutoff was used to estimate the associated sensitivity and specificity values. Positive and negative likelihood ratios are calculated from the sensitivity and specificity output. The data were analyzed using MetaboAnalyst 3.5 maintained by Xia Lab at McGill University. The statistical approach outlined here (Figure 1) was designed to provide meaningful and validated results optimized for the small sample size of the cohort, aiming to find potential biomarkers of GVHD to support future studies. To obtain an adequately powered sample size, patients who did not experience GVHD following an allograft were combined with patients who underwent an autologous SCT in this pilot project.

## 3. Results

Fourteen patients were analyzed in this study (Table 1). Of these, 10 underwent a myeloablative allograft and 4 underwent an autologous SCT. The entire study cohort was composed of three HLA-matched related donors, seven HLA-matched unrelated donors, and four autologous stem cell transplant recipients, with mean age of 50 (±10) years old; 57% patients were women, and 11 were Caucasians and 3 were African Americans.

Following preprocessing and filtering to remove low-quality data, in the individual patient datasets, the final aggregate, analyzable dataset consisted of 225 plasma lipids and 139 non lipid small-molecule metabolites derived from the patients.

### 3.1. Pre-Transplant Plasma Lipid and Metabolite Profiles Reveals Class Separation between Those Patients Who Ultimately Developed GVHD and Those Who Did Not

To estimate the potential of the lipids and metabolites to predispose SCT recipients to the development of alloreactivity in the form of either acute or chronic GVHD, post-conditioning and immediate pre-transplant plasma lipid and metabolite data were analyzed via PLS-DA. The degree of separation of patients with future GVHD against patients with no GVHD was visualized by the scores plot of the two principal components. The distance of class separation suggests that metabolic variation may correlate with the development of GVHD in patients undergoing SCT (Figure 2).

The top 20 variables that contributed to the separation observed in the PLS-DA model were represented by 2 metabolites and 18 lipids (Table 2). These characterized the metabolic variation present in the patients, pre-transplant, and correlates strongly with GVHD development post-transplant. Compared to patients with no GVHD, GVHD patients had decreased levels of 2-aminobutyric acid, hexose, monounsaturated fatty acids (14:1, 16:1, 18:1, 19:1, and 20:1), and poly-unsaturated fatty acid (20:3), as well as plasmenyl-ethanolamine (PE(p-34:1) or PE(o-34:2)). Further, GVHD patients presented elevation of monoacylglycerols (1-monoolein and 1-monopalmitin), diacylglycerols (38:5 and 38:6), along with elevated lysophosphocholine (14:0, 20:0), phosphocholines (28:0, 14:0/16:1, 16:0/18:3), and phosphoethanolamines (16:0/18:1, 18:0/22:5).

### 3.2. The More Important Variables for Class Separation Suggest Metabolic Pathway Tendencies Predispoising to Alloreactivity

So far, our study has identified 20 metabolites, the presence of which correlates with the eventual development of GVHD, suggesting an inherent metabolic disturbance that predisposes a patient towards alloreactivity as early as the day of SCT. Further examination of these metabolites indicates that they modulate three related metabolic pathways. Activated phospholipid metabolism appears to be one of the main alterations associated with GVHD pre-transplant. 1-monopalmitin is a monoacylglycerol that had a high VIP score (2.54), and its elevation in the GVHD group indicates phospholipid degradation in cell membranes to produce diacylglycerol, the precursor for MAGs. Alternatively, the non-GVHD state appears to be associated with elevation of monounsaturated and polyunsaturated fatty acids (Figure 3A). Hexose, more commonly called glucose, also had a high VIP score (2.67), indicating that elevated glucose uptake is on demand for energy production in the tricarboxylic acid cycle (TCA), as well as increased aerobic glycolysis required for hematopoietic cell proliferation. These processes increase the production of the reduced form of nicotinamide adenine dinucleotide (NADH) used in the electron transport chain, but its upregulation induces excessive reducing power that triggers processes such as fatty acid unsaturation and ROS production (Figure 3B). The levels of oxidative stress originated by excessive ROS production is controlled by the glutathione metabolism, where NADH is also used to reduce glutathione for its antioxidant action over ROS. The excessive demand for antioxidative process can deplete glutathione and its precursor cysteine, increasing the demand of 2-aminobutyric acid that can either modulate the glutathione synthesis or be used in ophtalmate synthesis, a tripeptide analog of glutathione, with similar compensative antioxidative actions (Figure 3C).

### 3.3. The More Important Variables for Class Separation Can Be Used to Build Models for GVHD Association

To evaluate the potential to build models to possibly predict future GVHD, the top 20 highest VIP scores were used in an exploratory analysis. Hence, the analysis showed that plasma metabolites and lipids obtained post-conditioning on day 0 prior to SCT, may be used to build predictive models for GVHD (Figure 4). Models with 2, 3, or 5 variables demonstrated the same level of performance, fitting the selection criteria, with AUCROC ranging from 0.915 to 0.935. With the criteria of finding the lowest number of predictors that can also physiologically explain the metabolic profile of the classes, a model with five predictors was chosen for further exploration as potential biomarkers for risk stratification of patients with the potential for development of GVHD following SCT.

### 3.4. Univariate ROC Curve Analysis Finds Potential Biomarkers of GVHD With Plasmatic Data Pre-Transplant

The optimal cutoff found for each predictor from the ROC curve was used to estimate the sensitivity and specificity, and to calculate the positive and negative likelihood ratios.

The five best biomarkers and their respective estimates are depicted in Table 3, and the comparison plot for each potential biomarker is presented in Figure 5 where presence of outliers is of notice. These data support the previous finding that a model with five metabolic biomarkers may provide a robust model for GVHD prediction.

All potential biomarkers show high sensitivity and specificity, except for FA 20:1, which, while it has maximum sensitivity, has low specificity. The positive likelihood ratio of 1-monopalmitin and DG (38:5) and DG (38:6) shows that a patient with a positive test result for these biomarkers will have very high odds of developing GVHD. 1-monopalmitin, 2-aminobutyric acid, and DG 38:5 showed CI ranging inside an acceptable AUROC values (0.6–1.0). Yet, the bootstrap confidence interval for DG 36:6 and FA 20:1 are indicative that these two compounds must be taken with caution despite their high sensitivity estimate.

The evaluation of SVM model performance analyzed by AUROC shows the five specific biomarkers model appears to be accurate in predicting the future development of GVHD (AUC = 0.995) with patient’s day 0 plasma drawn pre-transplant (Figure 6A). The predictive power of class probability of the SVM method was also tested and the confusion matrix represented by the probability plot in Figure 6B shows the method is robust without any misclassification.

## 4. Discussion

The identification of biomarkers that might enable the risk stratification of SCT patients with respect to the potential for development of GVHD has significant clinical utility in the execution of SCT protocols [28,29,30]. We have previously demonstrated that combined lipidomics and metabolomics approached are extremely useful in identifying putative metabolic biomarkers of disease [31]. Here, we demonstrate the discovery of 18 lipids and 2 metabolites separating the GVHD from the non-GVHD cohorts, also implicating specific metabolic pathways involved in the pathogenesis of alloreactivity.

2-aminobutyric acid is a byproduct in the cysteine biosynthesis pathway and relates to glutathione (GSH) metabolism. The presence of an alpha-amino acid as a predictor of GVHD after SCT is substantiated by the reported role of amino acid metabolic changes pre-transplantation [32]. Moreover, Reikvam et al. [33] showed that the altered metabolism of branched chain amino acids, as well as of isobutyryl-carnitine and propyonyl-carnitinine in lipids, before pre-conditioning are associated with the development of acute GVHD. Our results suggest that even as early as after conditioning therapy and before transplantation, the effects of amino acid and lipid metabolism may be associated with GVHD onset.

The metabolic implication of 2-aminobutyric acid in modulation of GSH homeostasis by production of ophthalmic acid is also well known [34]. Ophtalmate is an analog of GSH in which the cysteine group is replaced by L-2-aminobutyrate. It has been proposed that oxidative stress leads to intracellular depletion of GSH, depletion of cysteine, and consequent activation of ophthalmate synthesis [35]. 2-aminobutiric acid increases intracellular GSH levels by regulation of AMP-activated protein kinase and increase of the reduced form of nicotinamide adenine dinucleotide phosphate (NADPH). Moreover, it has been demonstrated that chemotherapy agents are capable of inactivating glutathione reductase, the enzyme that catalyzes the reduction of glutathione disulfide to the sulfhydryl form glutathione [36,37].

Particularly in the immune system, activated T cells undergoing clonal expansion have increased energy demands that increase production of ROS by the mitochondrial electron transport chain [38]. GSH is not necessary for cell activation, but activated T cells regulate their oxidative stress by using GSH, a key component for metabolic reprograming for cell differentiation [39]. Also, TCR ligation and binding with costimulatory molecules induces metabolic remodeling of the naive T cell to anabolic growth and biomass accumulation, and increases aerobic glycolysis [40].

A novel observation from the metabolic profile of GVHD-prone patients in our study is that monoacylglycerol (MAG), diacylglycerol (DAG), fatty acids, phospholipids, and plasmalogens metabolism are significantly altered, identifying lipids as potential mediators of GVHD. Lipid modulation is expected due to metabolic demands of compromised hematopoietic tissue, immunologic response, and underlined inflammatory profile of the patients related to the conditioning regimen. The effects of MUFA and PUFA from the omega-3 family in decreasing inflammation have been extensively studied in efforts to introduce intake of dietary lipids toward treatment of several diseases [41]. These lipids have the potential to decrease production of cytokines in response to LPS and increase the concentration of the anti-inflammatory cytokine IL-10 [42]. The ratio of saturated to monounsaturated fatty acids in membrane phospholipids is critical to normal cellular function. Alterations in this ratio have been correlated with cancer, and the oxidative stress characteristic of the pathology [43,44].

In a study analyzing plasma phospholipids changes in patients with acute leukemia, it was demonstrated that all phospholipids’ concentrations found in patients at the time of diagnosis were significantly lower than in the reference group [45]. Endogenous lipids are important not only in regulation of inflammation, but also in expressing antitumor functions in several types of cancer [46,47,48]. Free fatty acids may be esterified in cell membrane phospholipids undergoing hydrolysis by phospholipases to generate bioactive lipid mediators, including lysophosphocholine (LPC), diacylglycerol (DAG), monoacylglycerol (MAG), and unsaturated fatty acids (MUFA and PUFA). LPC is known to exert immune-regulatory activity by increases in the numbers of T cells, monocytes and neutrophils, and also inducing protein kinase C activation in bone marrow derived mast cells [45].

DAGs are not only a precursor for free fatty acids but also an important signaling molecule in cells. Protein kinase C (PKC) is the major cellular target of DAG. The protein kinase D (PKD) is a substrate of PKC responsible for several cell responses as proliferation, differentiation, apoptosis, and immune response through TCR signaling [49]. Under oxidative stress, ROS induces activation of PKD to protect the cell from oxidative-stress-induced cell death [50].

1-monopalmitin is a MAG formed via release of a fatty acid from DAG by diacylglycerol lipase. The hydrolysis of MAG to FFA and glycerol is conducted by the MAG lipase (MAGL) in different tissues, although ABHD6, a MAG hydrolase, has also been implicated in the pathogenesis of metabolic syndrome [51], inflammation [52], and in cancer [53]. Deletion or inhibition of ABHD6 activity has been shown to be beneficial in certain cancers [54]. The importance of 1-monoacylglycerols with a saturated fatty acid group is demonstrated by its accumulation upon ABHD6 suppression, and direct binding to the ligand binding domain of the peroxisome proliferator-activated receptors PPARα and PPARγ, and activating these transcription factors [55]. MAGs metabolism is also related to the effects of endocannabinoids in the immune system. The endocannabinoid receptor CB2 is identified as a peripheral receptor preferably present in B cells, T cells, macrophages, monocytes, natural killers, and polymorphonuclear. The enzyme MAGL that hydrolyzes MAGs also hydrolyzes 2-arachidonoylglycerol (2-AG), an endogenous endocannabinoid acting through CB2 receptors in the immune system with immune suppression effects [56,57].

## 5. Limitations

The small size of the cohort used in this study is a limitation that prevents definitive confirmation that the biomarkers identified in this study may be used as predictors to risk stratifying patients with respect to the potential for the development of GVHD. The small sample size also compelled us to combine patients with acute and chronic GVHD, to look at cumulative GVHD incidence and to pool autologous SCT controls with those allograft recipients who did not experience GVHD. Despite these limitations, this study allows an understanding of the metabolic milieu in the study population at the time of transplantation and can be used to direct future studies similar to other studies described in literature [58]. Furthermore, significance following the use of the statistical strategy to overcome the small sample size using Monte Carlo cross validation with bootstrap resampling to create confidence intervals further indicate the study results are of significance. Such approaches have been used previously in the literature as appropriate tools to deal with the small sample size problem [59].

## 6. Conclusions

Our study demonstrates that the pre-transplant lipidome and the metabolome of SCT recipients has significant potential towards their risk stratification with respect to the development of GVHD, indicating potential use as biomarkers for this purpose. The identified potential biomarkers indicate a pro-inflammatory metabolic profile in patients that will eventually develop GVHD. The role of GSH and its association with 2-aminobutyric acid, coupled with signs of altered glucose metabolism, support the hypothesis that in patients that will develop GVHD, GSH levels are excessively depleted due to elevated oxidative stress related to glucose metabolism, in response to chemotherapy treatment. In this scenario, both groups have elevated 2-aminobutyric acid as a compensatory mechanism, although in patients susceptible to develop GVHD, it is not produced in a large enough amount to compensate for the increased antioxidant demand. Furthermore, the decreased levels of plasma hexose indicate excessive glucose uptake for glycolysis, oxidative phosphorylation, and consequent ROS production. The protective effect of MUFA and PUFA as anti-inflammatory agents is decreased in patients that will develop GVHD, and elevation of phospholipids and DAG and MAG indicates an increased traffic of inflammatory lipids. This pro-inflammatory profile in patients with risk of GVHD is associated with immune suppression, and characterize an unfavorable environment coupled with the overwhelming physiologic impact of undergoing the donor graft, predisposing the host to GVHD.

## Figures and Tables

**Figure 1 cancers-11-01051-f001:**
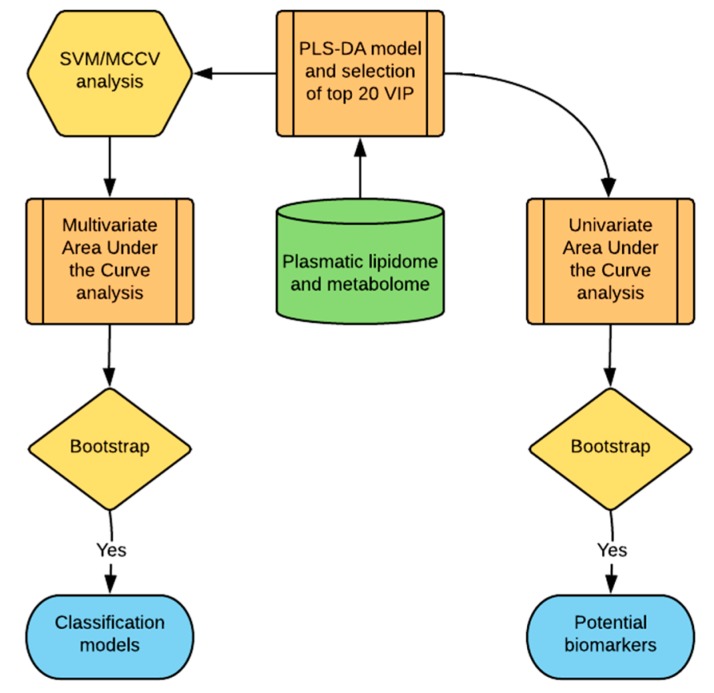
Statistical approach used for the identification of potential lipid and metabolite-based biomarkers for the prediction of the alloreactivity following stem cell transplantation (SCT). A supervised statistical approach in the form of partial least squares discriminant analysis (PLS-DA) was applied to the consolidated metabolomic and lipidomic data as a first step to find potential biomarkers and to detect the presence of class separation, if any, between graft vs. host disease (GVHD) and non-GVHD patients. The top 20 most important variables, which separates the groups, were selected by the variable importance in projection (VIP). A support vector machine (SVM) used these VIP variables to further find the binary classification of patients in the two groups. The result was cross-validated with Monte Carlo cross-validation (MCCV) and was used in a multivariate area under the curve of the receiver operating characteristic (AUCROC) analysis to select the best model based on low dimensionality and high accuracy. Model estimates were validated with bootstrap CI. The best model was used as reference to select potential biomarkers of GVHD through univariate analysis of the top 20 VIP that passed the criteria of high AUCROC estimate and low *t*-test *p*-value, and validated with bootstrap CI.

**Figure 2 cancers-11-01051-f002:**
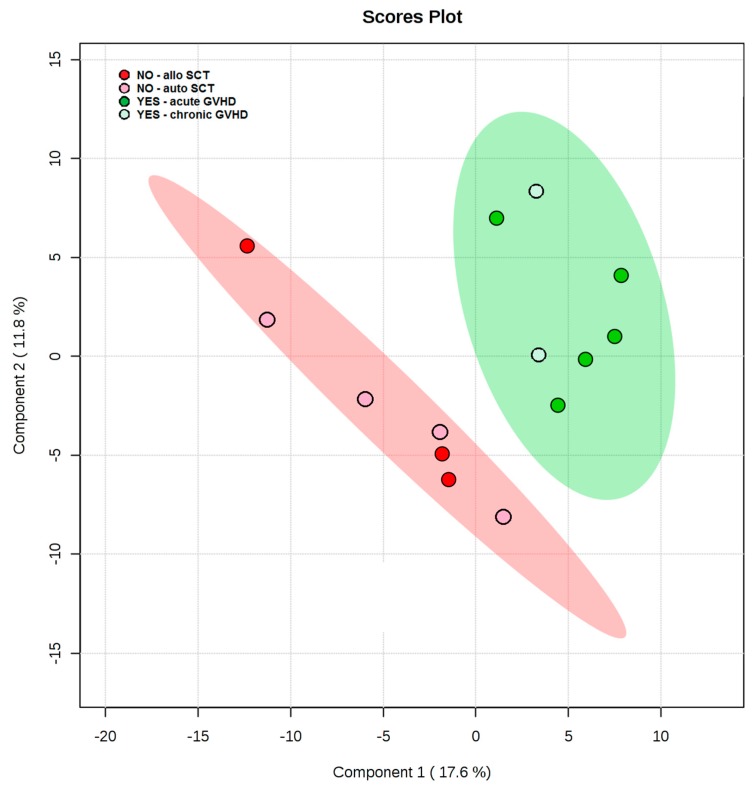
Supervised method PLS-DA identified the direction of maximum covariance between the data et and the class membership, extracting features in the form of latent variables. The model separation of GVHD (YES) and no-GVHD (NO) pre-transplant indicates metabolic differences for prediction of GVHD phenomena with no overlap in the 95% confidence intervals (shaded areas). In the NO group, the allogeneic donor type (allo SCT) is indicated by the red circle, and the autologous donor type (auto SCT) is indicated by pink circles. In the YES group, acute GVHD is indicated by the dark green circle, and the chronic GVHD is indicated by the light green circle.

**Figure 3 cancers-11-01051-f003:**
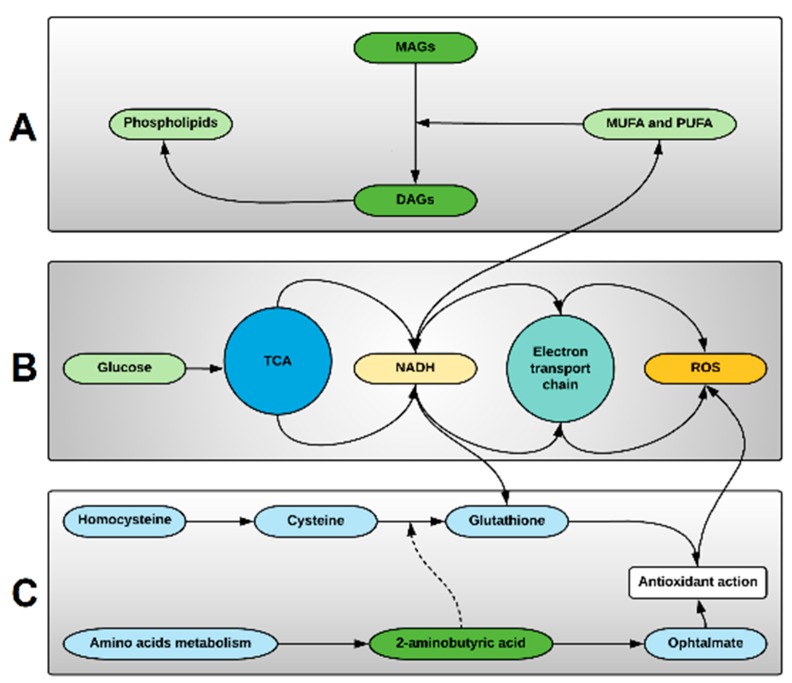
The more important lipids and metabolites associated with GVHD may indicate the main compensatory metabolic pathways modulated in hematopoietic cells pre-transplant. (**A**) Phospholipids metabolism are affected suggesting Phospholipase C activity to produce diacylglycerols (DAGs), and diacylglycerol lipase and monoacylglycerol lipase activity to produce monoacylglycerols (MAGs) and free fatty acids (FFA), respectively. Predominance of monounsaturated (MUFA) and polyunsaturated (PUFA) free fatty acids are result of increased fatty acyl-CoA desaturases activity, increasing the anticancer activity of ω-3 fatty acids. This metabolic pathway is linked to elevated levels of the reduced form of nicotinamide adenine dinucleotide (NADH), produced in the tri-carboxylic acid cycle (TCA), and important on desaturases activity. (**B**) Glucose uptake is increased due to increased energy demands in hematopoietic cells, and increased aerobic glycolysis with increased NADH, providing NADH for the electron transport chain and resulting in reactive oxygen species (ROS). (**C**) Oxidative stress compensation is achieved by activation of cysteine-glutathione pathway, and antioxidant action. Exacerbation of these mechanism leads to depletion of cysteine and glutathione, causing increased 2-aminobutyric acid-ophtalmate compensatory pathway activation originated from amino acids metabolism.

**Figure 4 cancers-11-01051-f004:**
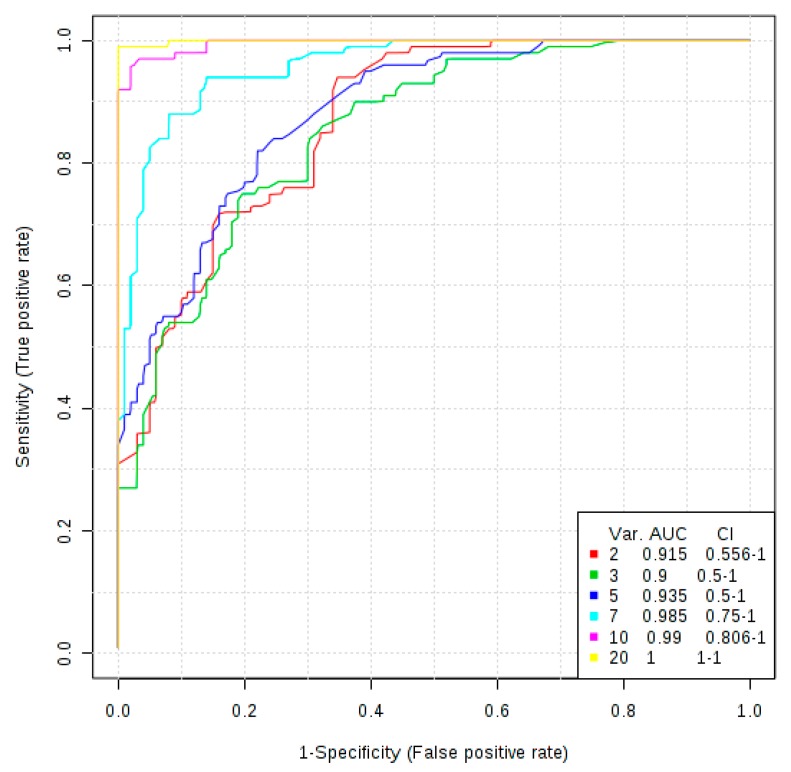
The top 20 variables from PLS-DA method were used to build appropriate classification models with their respective confidence interval. Predictive models pre-transplant demonstrated robust classification even with only two variables.

**Figure 5 cancers-11-01051-f005:**
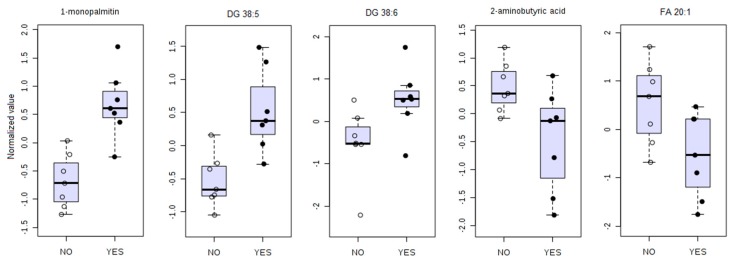
Comparison of five biomarkers of GVHD (YES) against no GVHD (NO). The comparative analysis revealed elevated 1-monopalmitin, DG (38:5), and DG (38:6) and decreased 2-aminobutyric acid and FA (20:1) in patients that will develop GVHD.

**Figure 6 cancers-11-01051-f006:**
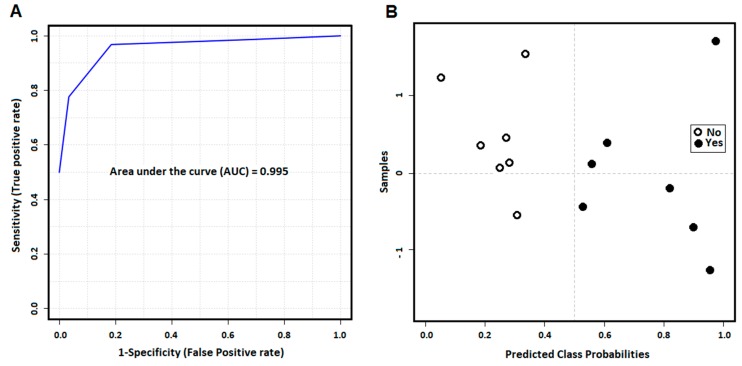
Final model with five specific biomarkers predicting GVHD with cross validation shows strong performance accuracy to predict GVHD (**A**). The cross-validation for the SVM method for model building and performance evaluation is performed multiple times to test a model with five specific biomarkers and it shows that the model can predict GVHD patients without misclassification (**B**).

**Table 1 cancers-11-01051-t001:** Demographic data for the cohort study.

Donor Type	Age	Gender	Race	BMI	Prep Regimen; GVHD Prophylaxis	Disease	GVHD (Grade/Severity)	GVHD Organs/System
**MRD**	60	Female	C	21.41	Bu/Cy; Cyls/MTX	MDS	Acute (1)	Skin, liver, GI
**URD**	50	Female	C	24.42	Bu/Cy; Tac/MMF	AML *	Acute (4)	Skin, GI
**URD**	31	Female	C	34.99	Bu/Cy; Tac/MTX	AML	Chronic (mod)	Skin, GI
**URD**	40	Female	C	42.08	TBI/Cy; Tac/MTX	ALL	Acute (2)	GI
**URD**	57	Male	C	22.09	Bu/Flu; Tac/MMF	MDS	Acute (2)	Skin
**URD**	50	Male	C	26.13	Flu/Mel; Tac/MTX	ALL	Acute (1)	Skin
**MRD**	59	Male	C	21.80	Bu/Flu; Cyls/MMF	MDS	Chronic (mod)	Oral, GI
**URD**	50	Female	C	28.85	Bu/Cy; Tac/MTX	CML ^&^	no	-
**MRD**	63	Female	C	27.81	Bu/Flu; Cyls/MTX	ET ^#^	no	-
**URD**	66	Male	C	29.45	Flu/Mel; Tac/MMF	MCL	no	-
**auto**	48	Female	C	17.60	BEAM; n/a	PTCL	n/a	-
**auto**	35	Female	AA	25.04	Mel 200; n/a	MM	n/a	-
**auto**	44	Male	AA	45.22	BEAM; n/a	NHL	n/a	-
**auto**	40	Male	AA	51.22	BEAM; n/a	ALL	n/a	-

BMI = Body Mass Index; MRD = match related donor; URD = unrelated donor; auto = autologous; C = Caucasian; AA = African American; HTN = hypertension; MI = myocardial infarction; n/a = not applicable. Bu = Busulfan; Cy = Cytoxan; Cyls = cyclosporine; MTX = methotrexate; Tac = tacrolimus; MMF = mycopehnolate mofetil; TBI = total body irradiation; Flu = fludarabine; Mel = melphalan; BEAM = carmustine, etoposide, cytarabine and melphalan; MDS = myelodysplasia; AML = acute myeloid leukemia; HL = Hodgkin lymphoma; CML = chronic myeloid leukemia; ET = essential thrombocythemia; MCL = mantle cell lymphoma; PTCL = Peripheral T-cell lymphoma NOS; MM = multiple myeloma; NHL = non-Hodgkin lymphoma; ALL = acute lymphoblastic leukemia; GI= gastrointestinal system * Secondary to chemotherapy; ^&^ blast crisis; ^#^ w/progression to myelofibrosis.

**Table 2 cancers-11-01051-t002:** The metabolites and lipids constituting the top 20 VIP’s that predict class separation on the day of transplant between those patients who eventually went on to develop GVHD as opposed to those who did not develop GVHD following SCT. Patients that went onto develop GVHD had decreased 2-aminobutyric acid, hexose, unsaturated fatty acids, and plasmenyl-ethanolamine PE (p-34:1) or PE (o-34:2), along with elevated monoacylglycerols and diacylglycerols, lysophosphocholines, phosphocholines. and phosphoethanolamines. GVHD’ patients (YES) and non-GVHD (NO) are represented by dark grey (high) or light grey (low), respectively.

Class	Predictors	VIP	GVHD	
			NO	YES
**Alpha-amino acid**	2-aminobutyric acid	1.80		
**Monosaccharide**	Hexose	2.67		
**Monoacylglycerol**	1-monoolein	2.11		
	1-monopalmitin	2.54		
**Diacylglycerol**	DG 38:5	1.97		
	DG 38:6	1.91		
**Fatty acid**	FA 14:1	1.90		
	FA 16:1	2.08		
	FA 18:1	1.92		
	FA 19:1	2.09		
	FA 20:1	2.01		
	FA 20:3	2.07		
**Lysophosphatidylcholine**	LPC 14:0	2.79		
	LPC 20:0	2.46		
**Phosphatidylcholine**	PC 28:0	2.29		
	PC 14:0/16:1	1.95		
	PC 16:0/18:3	1.95		
**Phosphatidylethanolamine**	PE 16:0/18:1	2.02		
	PE 18:0/22:5	2.23		
**Plasmenyl-ethanolamine**	PE (p-34:1) or PE (o-34:2)	1.75		

**Table 3 cancers-11-01051-t003:** Potential biomarkers performance for a model with top five predictors based on accuracy performance (area under the curve (AUC) estimate).

Predictor	AUC	*p*-Value	Sensitivity	Specificity	LR+	LR−
1-monopalmitin	0.96 (0.82–1.00)	0.0005	0.86 (0.71–1.00)	1.00 (1.00–1.00)	infinity	0.14
DG 38:5	0.96 (0.80–1.00)	0.003	0.86 (0.57–1.00)	0.86 (0.64–1.00)	6.0	0.17
DG 38:6	0.86 (0.57–1.00)	0.035	0.86 (0.43–1.00)	0.86 (0.64–1.00)	6.0	0.17
2-aminobutyric acid	0.86 (0.61–1.00)	0.029	0.86 (0.57–1.00)	0.71 (0.29–1.00)	3.0	0.20
FA 20:1	0.82 (0.49–0.97)	0.039	1.00 (1.00–1.00)	0.57 (0.29–0.86)	2.3	0.00

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
