# Peer review of "A Preliminary Investigation towards the Risk Stratification of Allogeneic Stem Cell Recipients with Respect to the Potential for Development of GVHD via Their Pre-Transplant Plasma Lipid and Metabolic Signature"

_cancers, 2019, doi:10.3390/cancers11081051_

Round 1
Reviewer 1 Report
The authors showed that the metabolic and lipid profiles prior to allogeneic hematopoietic stem cell transplantation (allo SCT) are closely related to the development of graft-versus-host disease (GVHD) in this pilot study. Since the sutdy was conducted in the small size and patients who had various backgrounds (i.e., donor source, conditioning regimen, and immunoprophylaxis) were enrolled, the definite conclusion cannot be drawn. However, they showed new findings that may be useful to control GHVD afer allo SCT.
Specific comments
The authors should describe the disease types and stem cell sources in Table 1.
The authors combined patients who did not experience GVHD following allo SCT and patients who underwent an autologous SCT in Fig.2 because of a shortage of enough samples. However, these patients were basically different in their immune status. The authors shoud indicate which one is which in Fig. 2.
The discussiion should be more concisely presented.
4. In the present study, acute and chronic GVHD were developed in 5
and 2 patients, respectively. Acute and chronic GVHD are thought to be differed in each other in terms of the pathophysiology. The authors
discuss on this point, considering the present findings.
5. The Discussion section should be more concisely presented.
Minor comments
The authors decribe GLUT1 in line 67、STAT3 and mTOR in line 82, NADH in line 297 and Fig. 3, NADPH in line 403, and PPAR in line 480 as the full spelling.
"Mastcells" in line 462 is mast cells.
In line 379, "The metabolic profile prior to bone marrow transplant" should be "The metabolic profile prior to SCT".
Reviewer 2 Report
This is an interesting study to analyze lipidome and metabolome regarding hematopoietic stem cell transplant and to identify biomarkers that predict GVHD development. Although the number of cases being analyzed is small, it is considered that the manuscript can contain important findings that will give hints to future researches. However, there are some issues to be clarified.
Major points
1. The authors' definition of GVHD (YES) seems vague. Is G1-4 acute GVHD or chronic GVHD defined as GVHD (YES)? It seems usual to consider G2-4 acute GVHD or chronic GVHD as clinically significant GVHD because G1 acute GVHD often does not require treatment.
2. Treatment before transplant may differ between allo-transplant and auto-transplant. Accordingly, were any differences in the lipid an metabolic signatures between the two groups before transplant?
3. Were there any differences in lipid and metabolic signatures according to acute GVHD, chronic GVHD and their target organs?
4. What are the functional implications of the identified lipid and metabolic biomarkers?
Reviewer 3 Report
This study was a small case series designed to evaluate changes of pro-inflammatory metabolic signature. Only 14 patients were accrued with myeloablative allograft in 10 cases and autologous in 4. The study identified 20 metabolites whose differential presence correlates strongly with the eventual onset of GVHD suggesting an inherent metabolic disturbance that predispose a patient towards GVHD as early as the day of SCT.
This article was well written to cover the recent advancements that have been achieved in this rapidly expanding area of research. However, it had been widely known that patients who developed later acute GVHD especially showed altered pretransplant amino acid metabolism. The authors truly need to do more experiments to convince the reviewer and other readers the results in the paper are correct. Actually, I did not learn much from the theoretical side of the event. A major revision may be required.
Round 2
Reviewer 1 Report
The revised version of manuscript was improved by editing according to the reviewers' comments.
Reviewer 2 Report
The manuscript significantly improved after revision.
Reviewer 3 Report
Most of my concerns from my previous review had been addressed.